# Turner Syndrome: Ocular Manifestations and Considerations for Corneal Refractive Surgery

**DOI:** 10.3390/jcm11226853

**Published:** 2022-11-20

**Authors:** Majid Moshirfar, Mark T. Parsons, Chap-Kay Lau, Nicholas A. Chartrand, Yasmyne C. Ronquillo, Phillip C. Hoopes

**Affiliations:** 1Hoopes Vision Research Center, Hoopes Vision, Draper, UT 84020, USA; 2John A. Moran Eye Center, Department of Ophthalmology and Visual Sciences, University of Utah School of Medicine, Salt Lake City, UT 84132, USA; 3Utah Lions Eye Bank, Murray, UT 84107, USA; 4College of Medicine-Phoenix, University of Arizona, Phoenix, AZ 85004, USA

**Keywords:** Turner Syndrome, refractive surgery, LASIK, PRK, SMILE

## Abstract

Turner Syndrome (TS) is the most common sex chromosome abnormality in females and is associated with physical changes, hormone deficiencies, increased risk of autoimmune disease, and ocular complications. In this article, we review the main ocular findings associated with TS and discuss their significance for the patient considering refractive surgery. We also present four cases of TS to highlight the clinical findings that may be present in these patients. The most common ocular manifestations include refractive errors, strabismus, and amblyopia. Less commonly, patients with TS may present with keratoconus, cataracts, glaucoma, uveitis, or other disorders of the posterior segment. When considering corneal refractive surgery in a TS patient, clinicians should perform a thorough ocular history, ask patients about hormone therapy and autoimmune conditions, and pay particular attention to any of the associated ocular symptoms of TS.

## 1. Overview of Turner Syndrome

Turner Syndrome (TS) is characterized by the absence of a second X chromosome resulting in phenotypic females (45, XO), though variations of this karyotype exist. It is the most common sex chromosome abnormality in females with an incidence of 1 in 2000 to 1 in 2500 live births [1]. Diagnosis may occur prenatally or later in life, with an average age at diagnosis of 15 years old [2]. Clinical manifestations of TS are varied and include short stature, webbed neck, cubitus valgus, a “shield chest” with widely spaced nipples, hearing loss, and various cardiovascular complications and malformations [1]. Endocrine disorders are common, including ovarian insufficiency, low levels of growth hormone, and hypothyroidism [3]. The risk of autoimmune conditions is also doubled in TS patients, including Hashimoto’s thyroiditis, Celiac disease, and inflammatory bowel disease, among others [4].

Management of TS begins at birth and focuses on addressing the possible complications associated with the disease. In addition to regular screening for the numerous complications associated with TS, this typically includes growth hormone therapy to manage short stature, as well as estrogen therapy at age eleven or twelve to induce puberty. A progestin is later added to the estrogen regimen [3,5]. Growth hormone is usually stopped by the age of eleven or twelve, while estrogen-progestin hormone replacement therapy (HRT) is continued until the age of fifty or later to mimic the timing of normal estrogen levels through menopause [5].

## 2. Ophthalmologic Complications of Turner Syndrome

Though perhaps less well-recognized, over half of TS patients may have an ocular defect, with most of these resulting in impaired vision [6]. The most common impairments include refractive errors, strabismus, and amblyopia (Table 1). Less common manifestations include red-green color blindness, blue sclerae, and external abnormalities, such as epicanthal folds, ptosis, and hypertelorism [7,8]. Additionally, some reports indicate an association with congenital glaucoma, early onset cataracts, uveitis, and other abnormalities of the anterior segment such as keratoconus, limbal stem cell deficiency, and anterior lenticonus [9,10,11,12,13,14]. Some cases of posterior segment complications have also been reported, including retinal vascular changes and retinal detachment [8,15,16,17,18,19]. In contrast with other complications of TS, ocular manifestations do not seem to vary between different karyotypes of TS [6,20,21].

Refractive error represents the most common ocular abnormality in those with TS, with existing literature reporting a prevalence of 40–42% [8,21]. Estimates of the rates of specific types of refractive error vary, but most reports estimate hyperopia to be the most common (27–34% of TS patients), though myopia (12–13%) is not uncommon [6,8]. Though the overall rate of refractive error is similar to the general adult population [22], refractive errors in TS appear to begin at a younger age. In these cases, refractive errors tend to be larger and more often hyperopic than in individuals of a similar age without TS [6,21]. Additionally, one study reported rates of astigmatism in children with TS that were three times the rate of the background pediatric population (16% versus 4–5.5%) [6].

Hormonal deficits in TS may affect refractive stability and refractive error. Considering the evidence that changes in sex hormones affect corneal hydration, shape, and refraction [23,24,25], it is reasonable to consider a potential hormonal effect on vision among those with TS. In addition, growth hormone deficiency was associated with refractive error in a longitudinal study, suggesting that other hormones may play a role in refractive error in TS [26]. While it is clear that hormones affect the eye, it should be noted that there is no direct clinical evidence that hormone replacement therapy (HRT) influences rates of refractive error in individuals [27,28].

**Table 1 jcm-11-06853-t001:** Ocular abnormalities associated with Turner Syndrome (TS) as part of the eight-point eye exam [6,8,27].

Eight-Point Eye Exam	Abnormalities Associated with TS
Visual acuity	High refractive error
	Amblyopia
Pupils	-
Extraocular motility	Strabismus
Intraocular pressure	Glaucoma
Confrontation visual fields	-
External examination	Epicanthal folds
	Ptosis
	Hypertelorism
Slit-lamp examination	Cataracts
	Uveitis
	Keratoconus
Funduscopic examination	Vascular changes
	Uveitis
	Glaucoma

Strabismus is one of the most significant ocular complications of TS. This has been reported in 13–33% of those with TS, compared to a rate of 4% in the background population [29], and likely contributes to the high rate of amblyopia in this population (16–30%) [6,8,20,21]. Esotropia appears to be more common form of amblyopia, with reported rates ranging from 5.9–20% of total TS patients, compared to 1.2–9% for exotropia [6,8]. Notably, one study presented a cohort with a higher predominance of exotropia [21]. Awareness of the association of TS and strabismus is important for adequate detection and prevention of amblyopia in these individuals.

Some case reports suggest that TS is associated with anterior segment dysgenesis, leading to abnormalities such as congenital glaucoma, keratoconus, and anterior lenticonus [8]. A 1997 report described a series of three patients, between the ages of 13 and 27, with TS and bilateral keratoconus. All three were successfully treated with penetrating keratoplasty, and none had other ocular complications of TS [10]. A later report described another TS patient with bilateral keratoconus who underwent the same treatment [30]. No other cases of TS and Keratoconus were found in the literature. Two cases of limbal stem cell deficiency (LSCD) in TS were reported in 2014 [14]. Though the reports are rare, these corneal disorders would have a particular impact on the patient considering refractive surgery.

Regarding lenticonus, a single report in 1979 describes a 20-year-old patient with 45 XO karyotype who was found to have anterior lenticonus in addition to several systemic disorders associated with TS [11]. Other evidence suggests that TS may be related to both early onset cataracts and glaucoma [8,27,31,32]. Notably, a 2016 prospective case–control study identified a statistically significant increase in central corneal thickness (CCT) measurements of 31 TS patients compared to 67 age-matched patients without TS [33]. In addition to providing evidence of corneal abnormalities in TS, this result also has implications for the management of glaucoma, as the increased CCT could lead to artificially high intraocular pressure measurements [33].

While the exact mechanism of anterior segment dysgenesis has not been elucidated, one report postulated that the presence of two genetically different cell lines in mosaic TS variants may disrupt the development of the anterior segment [32]. In contrast, a more recent study found no relationship between TS karyotype and eye complications [6]. It should be noted, however, that their findings include a 4% rate of anterior segment abnormalities in the monosomy group (*n* = 2/50) compared to a 22.2% rate in the mosaicism group (*n* = 2/9).

We identified seven cases of uveitis associated with TS in the literature [12,13,17,34]. Four were cases of anterior uveitis, and the remaining three reported unspecified bilateral uveitis, sympathetic ophthalmia, and punctate inner choroidopathy, respectively. Four of the seven patients had an autoimmune disease (Crohn’s disease, juvenile seronegative arthritis, or psoriasis), and two had markers of autoimmune disease without a symptomatic or diagnosed condition. As previously mentioned, individuals with TS have twice the risk of autoimmune disease [4]. This association may have played a role in the cases of uveitis in patients with TS.

In addition to those found in the literature, we present four unpublished cases of TS seen at our clinic, each consistent with the previously mentioned ocular manifestations of TS (Table 2). In the first case, a 33-year-old woman presented for evaluation of unilateral keratoconus of the left eye and strabismus with amblyopia of the right eye. She reported a family history of keratoconus in her father. Additionally, we identified a possible case of punctate inner choroidopathy in the left eye. A second case was a 29-year-old woman requesting evaluation for refractive surgery. She had bilateral high myopia, keratoconus, multiple peripheral choroiditis, and an early cataract. In contrast with the first case, this patient had no family history of keratoconus. Medical history included TS, valvular heart disease, migraines, and hearing loss. Considering her ocular history, she was not considered a candidate for refractive surgery. In the third case, a 27-year-old woman presented for evaluation of dry eye. She reported a medical history of TS, hypothyroidism, ovarian failure, and valvular heart disease. Finally, the fourth case was a 38-year-old woman who presented for evaluation of dry eye, reporting a medical history of TS, inflammatory bowel disease, and adrenal insufficiency. Ocular history included dry eye, amblyopia, and high hyperopia previously treated with refractive lens exchange. Three of these four patients reported concurrent HRT.

Due to the small number of cases in the literature, it is unclear whether many of the ocular conditions discussed have a true association with TS or represent associations by chance. However, a recent population-based study analyzed the hospital inpatient and outpatient diagnostic codes of 1156 patients with TS and compared them with those from the background population. This analysis found an increased total risk of ocular disorders among those with TS [27]. Additionally, risk was increased for nearly every ophthalmologic condition, including corneal disorders, cataracts, glaucoma, strabismus, refractive errors, and disorders of the vitreous and retina. These findings support the association of TS with a wide range of ocular conditions despite the low number of cases in the literature.

## 3. Corneal Refractive Surgery in Turner Syndrome

The general criteria for refractive surgery patient selection include individuals 18 years or older, a stable refraction, corneal thickness of at least 500 µm [35], and an absence of any significant ocular pathology [36]. Patients are most often excluded from refractive surgery due to abnormal corneal topography or thin corneas [37]. Other notable reasons for exclusion include unstable refractive error, high refractive error, cataracts, glaucoma, or significant dry eye [38].

We identified no cases in the literature regarding the outcome of refractive surgery in TS patients. Considering the association of TS with ophthalmologic disorders, a patient with this syndrome who is considering refractive surgery should undergo a comprehensive ophthalmic examination to rule out any contraindicating conditions. They should be asked about refractive history and any current hormone replacement therapy (HRT) due to its potential effect on refractive stability. Ideally, same-day refraction could be compared to previous records to assure stability.

Despite sparse literature regarding HRT and refractive surgery, one study in 2006 compared the outcomes of LASIK surgery among a group of post-menopausal women on HRT, a group of women on oral contraceptives (OCP), and a control group [39]. After 2 months, the women on HRT had worse refractive outcomes than both the OCP and control groups. However, a previous study on PRK found that visual outcomes among premenopausal women on HRT were no worse than controls, while only the postmenopausal HRT group had worse outcomes [23]. Though the premenopausal HRT sample size was small (*n* = 7), these findings suggest that age may be a more important factor than HRT on surgical outcomes. Though these findings regarding HRT are mixed, we recommend assessing refractive stability with care.

Considering the high rates of strabismus and amblyopia in this population, the TS patient should also be assessed for these conditions. Strabismus and double vision are rare complications of refractive surgery, and patients with preexisting strabismus or significant anisometropia are at higher risk of strabismic deterioration [40]. However, some studies have reported that certain types of strabismus, especially hyperopic accommodative strabismus, can be improved with refractive surgery. Evaluation should include assessment of any prism in glasses, the type of strabismus, and stratification by risk of deterioration [41]. In cases of amblyopia, patient discussion should take into account the limited visual potential of the amblyopic eye.

As discussed, there is a potentially increased risk of corneal abnormalities such as keratoconus in those with TS. Even in cases without pre-existing keratoconus, these patients may carry a similarly increased risk of post-LASIK ectasia. Because of this, we recommend that corneal topography and pachymetry results be reviewed with a more sensitive threshold for abnormalities than in the general population. Notably, the average central corneal thickness in TS patients may be higher than typical [33]. It is unclear how this finding might relate to the risk of corneal disorders, but it should be kept in mind when interpreting corneal test results. Screening for cataracts and glaucoma should occur regardless of patient age, and interpretations of intraocular pressure measured by applanation tonometry should consider any increased central corneal thickness, if present [33].

Finally, due to their increased risk of autoimmune disease, patients with TS should be asked about any history of these conditions. Many systemic autoimmune diseases have traditionally been considered contraindications for refractive surgery due to high complication rates. More recently, some studies have suggested that the surgeries could be safer than previously thought in these patients [42,43], though the topic has been debated [44]. The decision should consider the stability of the condition, any recent uveitis or systemic flares, treatment with biologics, and the resultant state of immunocompetence. Other managing physicians should also be consulted, and patients should be aware of the potential associated risks. The increased prevalence of autoimmune disorders among patients with TS makes them a particularly important part of the patient history.

If the decision is made to proceed with refractive surgery in a TS patient, we recommend choosing the type of surgery based on the individual’s specific manifestations of the disease. In patients with autoimmune diseases, for example, LASIK is generally recommended over PRK, as it carries a lower risk of postoperative inflammation and scarring [42,44]. Both LASIK and PRK have been used in patients with strabismus [40,41]. Given that it is typically more complicated to carry out enhancements following PRK [45], LASIK may be preferable in patients on hormone replacement therapy (HRT), as these patients have a potentially increased risk of refractive regression following surgery [39]. PRK would be more appropriate in a patient with dry eye or concerns regarding corneal topography or thickness [45]. Though small-incision lenticule extraction (SMILE) is a potential treatment option, there is little literature on the use of SMILE in specific cases such as autoimmune disease and HRT. As the circumstances of each patient will vary, these factors should be weighed in order to make the most appropriate decision for the safety and outcomes of each individual case.

In summary, Turner Syndrome is associated with both systemic and ocular disorders that may affect the outcome and safety of corneal refractive surgery. High refractive errors, strabismus, corneal abnormalities, and autoimmune disease are particularly significant. Though TS patients may qualify for refractive surgery, eye care professionals should perform a thorough preoperative workup and have increased sensitivity to signs of these disorders. Postoperatively, patients should be followed closely to manage any concurrent ocular conditions or complications associated with refractive surgery.

## Figures and Tables

**Table 2 jcm-11-06853-t002:** Summary of Turner Syndrome cases.

Cases	1	2	3	4
Refractive	Myopia, high astigmatism OU Amblyopia OS	High Myopia OU	Myopic Astigmatism OU	High Hyperopia (s/p Refractive Lens Exchange) Amblyopia OS
Motility	Strabismus OS			Strabismus OS
Cornea	Keratoconus OS > OD	Keratoconus OU	Dry Eye Corneal injury OS	Dry Eye
Lens		Early cataract OD		
Fundus	Punctate Inner Choroidopathy OS	Peripheral Multiple Choroiditis OD		
Endocrine	Hypothyroidism	HRT	HRT	Hypothyroidism Adrenal Insufficiency HRT
Autoimmune				Crohn’s Disease
Other Medical History		Valvular Heart Disease Hearing Loss	Valvular Heart Disease Left Eardrum Repair	

Abbreviations: OD, right eye; OS, left eye; OU, both eyes; HRT, hormone replacement therapy; s/p, status post.

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
