# Peer review of "Turner Syndrome: Ocular Manifestations and Considerations for Corneal Refractive Surgery"

_jcm, 2022, doi:10.3390/jcm11226853_

Round 1

Reviewer 1 Report

This study provides a comprehensive review of Turner Syndrome, its main ocular associations and their significance for patients considering refractive surgery. The authors have also provided four case reports of TS to highlight some of the key clinical findings that manifest in patients with TS. I have a few comments and suggestions for improvement as follows:

1)    Page 2, Lines 71-76. The authors mention that strabismus is one of the most significant ocular complications of TS. Was there any one particular type (e.g. exotropia or esotropia) of strabismus that was more common when associated with TS? Would be nice to see the prevalence for these.

2)    Page 2, Line 77. The authors briefly mention an association of TS with congenital glaucoma, but don’t go into much detail about it beyond that. In particular, it would be good to elaborate further on how they are associated and the mechanisms underpinning this relationship between the two conditions.

3)    In Table 1, glaucoma is placed alongside intraocular pressure. However, given that glaucoma is defined as an optic neuropathy, I think it would be better placed under funduscopic examination as intraocular pressure is an independent risk factor for glaucoma but can’t be used as a sole metric for diagnosing glaucoma.

4)    Given that there is a large emphasis on refractive surgical considerations in patients with TS, are the authors able to provide some additional case studies on TS patients that have undergone refractive surgery and their outcomes?

5)    Given the vast number of possible ocular conditions that can be associated with TS in addition to multiple refractive surgical options, it would be nice to see some further insight into considerations around selection of the most suitable surgical option (e.g. LASIK, PRK, SMILE etc.) given a TS patient with a particular profile of associated ocular conditions. Perhaps a table highlighting various ocular conditions and the key surgical considerations would help supplement this too.

Reviewer 2 Report

This is a comprehensive work about the ocular manifestations in Turner Syndrome and their implications regarding corneal refractive surgery. Authors have made a thorough literature search, the report of findings is well organized and they provide 4 cases of their own clinic which add more information to the previous literature. The tables that authors provide are really helpful to meet the goals of the manuscript.

I only have few minor comments:

1. As the readers of this papers might probably be eye care professionals, they might not be familiarized with "HRT" abbreviation. The first time that "HRT" appears is in line 42, and the second time on line 71. Although it is a regular practice to specify the meaning of abbreviations the first time they appear, I do believe that writting "hormone replacement therapy (HRT)" on line 71 would be helpful for the reader. 

2. The authors make some insightful comments on section 3 about the considerations for corneal refractive surgery in TS. However, the title says "refractive surgery" so readers might think that other types of refractive surgeries are also considered in this paper. I would suggest to change the title to "... considerations for CORNEAL refractive surgery".

3. In line 180 "history of recent flares". Wouldn't be more appropriate to ask for "history of recent uveitits" or "history of recent uveitis flares"? (I guess that authors are not talking about dry eye flares due to context).

4. Line 188 "...opthalmologist should perform a thorough preoperative workup...". As preoperative evaluations can also be performed by optometrists, I suggest to write "opthalmologist and optometrist should...", or "eye care professionals should..."

Round 2

Reviewer 1 Report

I thank the authors for their careful consideration of my comments and suggestions, the manuscript has been substantially improved as a result.